# Robot-Assisted Renal Surgery with the New Hugo Ras System: Trocar Placement and Docking Settings

**DOI:** 10.3390/jpm13091372

**Published:** 2023-09-13

**Authors:** Francesco Prata, Gianluigi Raso, Alberto Ragusa, Andrea Iannuzzi, Francesco Tedesco, Loris Cacciatore, Angelo Civitella, Piergiorgio Tuzzolo, Giuseppe D’Addurno, Pasquale Callè, Salvatore Basile, Marco Fantozzi, Matteo Pira, Salvatore Mario Prata, Umberto Anceschi, Giuseppe Simone, Roberto Mario Scarpa, Rocco Papalia

**Affiliations:** 1Department of Urology, Fondazione Policlinico Universitario Campus Bio-Medico, 00128 Rome, Italy; gianluigi.raso@unicampus.it (G.R.); alberto.ragusa@unicampus.it (A.R.); andrea.iannuzzi@unicampus.it (A.I.); francesco.tedesco@unicampus.it (F.T.); loris.cacciatore@unicampus.it (L.C.); a.civitella@policlinicocampus.it (A.C.); p.tuzzolo@policlinicocampus.it (P.T.); giuseppe.daddurno@unicampus.it (G.D.); pasquale.calle@unicampus.it (P.C.); salvatore.basile@unicampus.it (S.B.); marco.fantozzi@unicampus.it (M.F.); matteo.pira@unicampus.it (M.P.); r.scarpa@policlinicocampus.it (R.M.S.); rocco.papalia@policlinicocampus.it (R.P.); 2Simple Operating Unit of Lower Urinary Tract Surgery, SS. Trinità Hospital, Sora, 03039 Frosinone, Italy; mario.prata@libero.it; 3Department of Urology, IRCCS “Regina Elena” National Cancer Institute, 00144 Rome, Italy; umberto.anceschi@ifo.it (U.A.); puldet@gmail.com (G.S.)

**Keywords:** docking, Hugo RAS, off clamp, robotic partial nephrectomy, trocar configuration

## Abstract

The current literature relating to the novel Hugo^TM^ RAS System lacks consistent data concerning the bedside features of robot-assisted partial nephrectomy (RAPN). To describe the trocar placement and docking settings for RAPN with a three-arm configuration to streamline the procedure with Hugo^TM^ RAS, between October 2022 and April 2023, twenty-five consecutive off-clamp RAPNs for renal tumors with the Hugo^TM^ RAS System were performed. We conceived a trouble-free three-arm setting to ease and standardize RAPN trocar placement and docking settings with Hugo^TM^ RAS. Perioperative data were collected. Post-operative complications were reported according to the Clavien–Dindo classification. The eGFR was calculated according to the CKD–EPI formula. Continuous variables were presented as the median and IQR, while frequencies were reported as categorical variables. Off-clamp RAPNs were successfully performed in all cases without the need for conversion or additional port placement. The median age and BMI were 69 years (IQR, 60–73) and 27.3 kg/m^2^ (IQR, 25.7–28.1), respectively. The median tumor size and R.E.N.A.L. score were 32.5 mm (IQR, 26–43.7) and 6 (IQR, 5–7), respectively. Two patients were affected by cT2 renal tumors. The median docking and console time were 5 (IQR, 5–6) and 90 min (IQR, 68–135.75 min), respectively, with slightly progressive improvements in the docking time achieved. No intraoperative complications occurred alongside clashes between instruments or with the bed assistant. In experienced hands, this simplified three-instrument configuration of the Hugo^TM^ RAS System for off-clamp RAPN resulted in feasible and safe practice, providing patient-tailored trocar placement and docking with non-inferior peri-perioperative outcomes to other robotic platforms.

## 1. Introduction

After the introduction of robotic platforms, urologists have been captivated by the idea of shifting all laparoscopic procedures to robotic surgery in order to overcome the intrinsic limitations of laparoscopy and to further push beyond the application of minimally invasive surgery [1,2,3,4]. Moreover, the last two decades have witnessed a significant evolution of robot-assisted surgery (RAS) to the point where it dominates minimally invasive scenarios in the majority of centers for many surgical procedures. Most urologists strongly support RAS, underlining how, in comparison to laparoscopy, surgical results are at a high standard with less chance of conversion to open surgery, lower blood loss, shorter operating times, and quicker discharge [5,6]. The technical advantages of RAS include improved wrist articulation for a broader range of motion, enhanced three-dimensional (3D) vision with magnification, tremor elimination, and surgeon comfort while operating from a remote console. These features contribute to reducing the learning curve (LC) of surgeons transitioning to minimally invasive surgery [7,8].

The pioneering robotic platform that was released in the early 2000s was the DaVinci Surgical System (Intuitive Surgical, Mountain View, CA, USA), and it is currently the market leader [9]. Despite its acknowledged benefits, the widespread adoption of this robotic system has been hindered by costs. Recently, new robotic platforms have been conceptualized with alternative technical features to improve the potential limitations of previous robotic systems and reduce procedural costs [10]. In this context, the introduction of the Hugo^TM^ RAS System has the broken robotic monopoly, aiming to enhance the diffusion of RAS technology without imposing a burden on the healthcare system. One of the main features of Hugo^TM^ RAS is the modularity of four independent arm-carts that allow more tailored surgery to be achieved, and thus, the possibility of adapting surgical strategies to unique cases [11,12]. Other technical advantages are represented by a larger working space for the bedside assistant, the use of more ergonomic trocar positioning, and the cost-effectiveness of single procedures. 

Partial nephrectomy stands as the gold standard treatment for localized renal tumors “whenever technically feasible” due to the reduced impairment of postoperative renal function and a lower theoretical risk of cardiovascular events [13]. The European Association of Urology (EAU) guidelines strongly recommend offering partial nephrectomy as the treatment of choice to T1 patients (maximum diameter < 7 cm). Nevertheless, T2 patients (>7 cm) could benefit from partial nephrectomy in the case of solitary kidney or chronic kidney disease if technically attainable [13]. Renal surgery, particularly nephron-sparing surgery (NSS), is one of the most challenging procedures in urology when considering the steep LC and the surgical experience needed [14]. Additionally, anatomic variability and tumor location could present obstacles when securing enucleation, potentially elevating the risk of peri-operative complications, such as bleeding or urinary leakage. Thus, many factors should be considered concerning the use of RAS for NSS before establishing the best surgical approach. 

In the context of minimally invasive NSS, robot-assisted partial nephrectomy (RAPN) has emerged as a prominent alternative to laparoscopy, allowing for improved peri-operative outcomes while displaying comparable oncological results [15,16]. The increasing adoption of NSS and the widespread embrace of the robotic platform by the urologic community has led to RAS largely replacing laparoscopic and open approaches in the setting of partial nephrectomy. Nevertheless, trocar placement and docking represent critical steps in robot-assisted renal surgery due to the need to standardize all phases of the procedure and enhance intra-operative outcomes [17,18]. On one hand, the introduction of RAPN, in comparison to laparoscopy, has expanded the application of NSS to more complex renal masses. On the other hand, the use of all four robotic arms, in addition to bed-assistant laparoscopic ports, increased the number of potential instrument collisions, compromising the safety of the procedure. Despite the initial enthusiasm for this innovative platform, instrument clashing represents an unsolved issue during renal surgery, even when closely adhering to Medtronic’s recommended port placement, potentially limiting the adoption of the Hugo^TM^ RAS system in straightforward standard cases [19,20]. Moreover, the paucity and fragmentation of the existing literature on the Hugo^TM^ RAS system hindered the provision of consistent and homogeneous data for the bedside features of this novel platform. Against this background, we aimed to describe our experience of trocar placement and docking settings for the largest series of RAPNs with the new Hugo^TM^ RAS System.

## 2. Materials and Methods

### 2.1. Patient Population

Between October 2022 and April 2023, 25 consecutive patients underwent off-clamp RAPN for renal tumors with the Hugo^TM^ RAS System at our institution: a high-volume center for off-clamp laparoscopic partial nephrectomy. The study included all patients who were eligible for RAPN. Patients were excluded if they displayed contraindications for partial nephrectomy: gross hematuria, not technically feasible, evidence of infiltration on conventional radiological imaging, and a clinical-stage > T2 (cT3-4). Written informed consent was obtained from all included patients. The baseline and perioperative data of patients were collected. Moreover, all subjects underwent pre-operative urine culture and imaging through a computed tomography (CT) scan. Renal masses were classified according to the R.E.N.A.L. score [21].

### 2.2. Endpoints, Data and Statistical Analysis

The primary endpoint of this study was to describe a three-arm trocar placement and docking setting for NSS using the Hugo^TM^ RAS System, specifically for off-clamp RAPN. Hugo^TM^ RAS was equipped with 11 mm and 8 mm trocars for endoscope and robotic instruments, respectively. Post-operative complications were reported according to the Clavien–Dindo classification [22]. Body mass index (BMI) was calculated as the weight in kilograms divided by height in meters, squared (kg/m^2^), and the estimated glomerular filtration rate (eGFR) was calculated according to the Chronic Kidney Disease Epidemiology Collaboration (CKD-EPI) formula. Continuous variables were presented as the median and interquartile ranges (IQRs), while frequencies were used to report categorical variables. STATA (StataCorp. 2021. Stata Statistical Software: Release 17. College Station, TX, USA: StataCorp LLC) was used for statistical analyses.

### 2.3. Trocar Placement and System Docking

After the induction of general anesthesia and trans-urethral bladder catheter positioning, patients were secured on lateral decubitus on the opposite side of the renal mass with the breakpoint of the operating table at the level of the last intercostal space. We used a modified extended flank position, placing the patient at the edge of the surgical bed and exploiting a moderate flexion of 45° in order to expand the operating space between the homolateral iliac spine and ribs margin. Through an open access technique, the first robotic trocar (11 mm, endoscope port) was placed trans-peritoneally along the pararectal line, approximately 14 cm far below the xifo-pubic line. Two additional robotic ports (8 mm, right- and left-hand instruments) were placed under view, at least 8 cm laterally from the camera port, observing a 2 cm safety margin from all bone prominences. Two more laparoscopic ports (12 mm) for the bed-assistant were positioned medially, about 8 cm from the robotic ports, in order to obviate unexpected clashes with robotic instruments (Figure 1 and Figure 2).

The position of the bedside assistant, either standing or seated, depends on the individual patient’s anatomical characteristics and the height of the surgical bed, as well as the docking and tilt angles of the robotic arms. The first surgeon needs to be seated during surgery for the optimal handling of Hugo^TM^ RAS controllers.

The suggested Hugo^TM^ RAS trocar configuration using Medtronic is based on 4 robotic arms, which include an 11 mm optic port and three 8 mm robotic instruments. This leaves enough space for only one bedside assistant laparoscopic trocar. In our surgical setup, we utilized three robotic arms in a three-instrument configuration and provided two 12 mm laparoscopic trocars for the bedside assistant. The rationale behind this trocar arrangement was to provide the assistant with a more active role during off-clamp partial nephrectomy. This involvement included utilizing two surgical suctions with irrigation at the same time. The simultaneous suction and irrigation of the resection bed offered the clear-cut visualization of tumor borders during enucleation and a precise discrimination between healthy renal parenchyma and the renal mass. Pneumoperitoneum was induced through the AirSeal^TM^ system (SurgiQuest, Milford, Connecticut, USA©), maintaining a standard intraabdominal pressure of 12 mmHg and a moderate increase to 15–20 mmHg during the off-clamp tumor enucleation. Only three arm carts (Figure 3) were used and placed behind the back of the patient, while the energy tower was positioned at the bottom of the bed.

Before docking, the arm carts were positioned 45 to 60 cm away from the patient and adjusted according to a new set-up arrangement.

We opted for a three-arm configuration to ensure generous bedside working space and prevent potential collisions between the robotic arms, both internally and externally, and with the bed assistant. According to the described setup, docking and tilt angles were displayed as follows:

**Right RAPN**

Docking angleTilt angleEndoscope275°−45°Right arm310°−15°Left arm225°−15°**Left RAPN**
Docking angleTilt angleEndoscope90°−35°Right arm135°−30°Left arm45°−45°

### 2.4. Surgical Procedure

The first surgeon, bed assistants, and scrub nurses taking part in the operations had all completed the technical training on the Hugo^TM^ RAS System provided by Medtronic at the ORSI Academy (Aalst, Belgium). All surgeries were performed using a traditional trans-peritoneal approach performed by a single surgeon with extensive experience in the off-clamp technique and minimally invasive partial nephrectomy. Specifically, the primary surgeon had over 10 years of experience and conducted more than 500 off-clamp laparoscopic partial nephrectomies. In addition, to gain confidence with the robotic system, following the certified training provided by Medtronic, an initial series of 15 robotic radical prostatectomies with the Hugo^TM^ RAS System were successfully completed.

Monopolar curved scissors were used for the right arm, switching to a large needle driver when suturing was required. The left robotic arm was equipped with Cadiere or fenestrated forceps. A 0° angle lens was utilized for the majority of procedures, while a 30° angle was the lens of choice in the case of posterior renal mass. After the paracolic gutter incision, peri-renal fat tissue was isolated, and the kidney was mobilized to expose the renal mass along with the surrounding parenchyma, enabling the clear identification of tumor borders. Mass enucleation was performed as an off-clamp RAPN following a clear-cut cleavage plane, managing bleeding from the resection bed through double suction and irrigation. Hemostasis was achieved with specific pin-point coagulation using monopolar energy. Renorraphy was performed in all cases with a 2/0 Monocryl single running suture secured with Hem-o-lok clips, using the sliding-clips technique. After a normotensive control, hemostatic agents (TachoSil^®^, TABOTAMP^TM^, or Floseal^®^) were applied on the resection bed to refine hemostasis. A single drain was introduced through the inferior 8 mm port, placed under direct vision, and Gerota’s fascia was closed.

## 3. Results

During the study period considered, 25 off-clamp RAPNs were performed successfully. The demographic patients’ characteristics are reported in Table 1.

The male/female ratio was 3.16. The median age and body mass index (BMI) were 69 years (IQR, 60–73) and 27.3 kg/m^2^ (IQR, 25.7–28.1), respectively. Eighteen patients were classified by the American Society of Anesthesiology (ASA) with a score of II (72%), and 14 suffered from blood hypertension (56%). Median preoperative hemoglobin, creatine, and eGFR were 15 g/dl (IQR, 13.8–15.5), 0.92 mg/dl (IQR, 0.81–1.07) and 84.6 mL/min/1.73 m^2^ (IQR, 64.5–90.9), respectively. The majority of patients had right renal masses (14, 56%), while 11 of them (44%) displayed left renal tumors. All patients were affected by single unilateral lesions, while only one patient had two homolateral renal tumors (4%). The median tumor size was 32.5 mm (IQR, 26–43.7). Twenty-three out of twenty-five patients in our cohort displayed cT1 renal masses, with the exception of two (8%) that were affected by cT2 renal tumors. The median R.E.N.A.L. score was 6 (IQR, 5–7).

Regarding intra-operative data, the median docking time was 5 min (IQR, 5–6). Specifically, there was a considerable improvement in the docking time from the first procedure to the last, reducing the time required from 10 to approximately 2 min. The median console time was 90 min (IQR, 68–135.75 min). The median estimated blood loss (EBL) was 175 mL (IQR, 100–400 mL). No intraoperative complications were recorded neither additional port placement was needed. There were no robotic instrument collisions within the abdomen and no clashing between the robotic arms and the bedside assistant ports. There were no unexpected technical failures in the system.

Postoperative data are reported in Table 2. The median hospital stay was 3 days (IQR, 3–4). Three patients (12%) presented post-operative Clavien–Dindo 2 complications: one developed a fever that was treated with intravenous antibiotic administration, while the other needed a single-unit blood transfusion due to postoperative anemia. No other complications occurred in our cohort. Before discharge, the median hemoglobin, creatine, and eGFR were 11.5 g/dl (IQR, 10.2–12.8), 0.9 mg/dl (IQR, 0.82–1.12), and 81.9 mL/min/1.73 m^2^ (IQR, 60.6–89.5), respectively.

Pathological reports showed 8 (32%) benign and 17 (68%) malignant lesions. This included one chromophobe (4%), two angiomyolipoma (8%), six oncocytoma (24%), ten clear cell carcinoma (40%), and six papillary carcinoma (24%). No positive surgical margins were reported at the final pathological evaluation.

## 4. Discussion

In the era of advancement in robotic surgery, alternative platforms with novel features have become available. The Hugo^TM^ RAS System currently stands out as the most comprehensive and promising alternative to the standard DaVinci system, especially in the field of minimally invasive urology. NSS has been widely recognized as the treatment of choice for the management of renal masses, and partial nephrectomy techniques have transitioned from an open to a minimally invasive approach [4,7,16]. Current international guidelines and the available evidence have strongly supported the use of RAS for partial nephrectomy as a compelling alternative to open surgery and laparoscopy for cT1 renal tumors [13]. As robotic technology becomes more accessible, the number of RAPNs is increasing exponentially, and contemporary series demonstrated their efficacy and viability even for high-nephrometry score renal tumors [1,4,23]. However, even if guidelines suggest that partial nephrectomy should be favored over radical treatment “whenever technically feasible”, RAPN for complex masses can be associated with a higher peri-operative complication rate, and RAS is not far from being widely adopted for high-nephrometry renal tumors. Nevertheless, the majority of the present evidence is based on previous-generation robotic platforms, and further studies on renovated ergonomic robotic systems are eagerly awaited.

The incorporation of novel technologies is likely to influence how urologist approach partial nephrectomy. The ongoing pursuit of tailored surgical solutions seems to find a natural fit with the Hugo^TM^ RAS platform due to its modularity and ergonomic features [11,12,19,20]. Challenging procedures, such as anatomical complexity, tumor location or nearness to the renal sinus, and potential arms clashing, make RAPN demanding for more ergonomic systems. Hugo^TM^ RAS demonstrated improved modularity through separate arm carts, which could either reduce the docking time or reduce the intraoperative rate of uneventful collisions between robotic and laparoscopic instruments. These technical nuances may be crucial in the context of RAPN, where the coaction between the master surgeon and the bedside assistant is a critical factor in preventing significant intraoperative bleeding. Moreover, the Hugo^TM^ RAS independent-cart docking system offers a real-time, customizable workspace for the bedside assistant. This adaptability is particularly valuable for patients with complex conditions, such as prior abdominal surgery, increased BMI, tumor location and complexity, or renal anatomic variability, in which an off-clamp approach could intimidate even experienced surgeons.

The Hugo^TM^ RAS System, developed by Medtronic, offers a wide range of robotic instruments with both mono- and bipolar energy capabilities. These instruments are designed to assist surgeons when performing various types of procedures and to adapt to different surgical scenarios. The 8 mm instruments currently available for robotic arms include: Monopolar Curved Shears, Bipolar Fenestrated Grasper, Bipolar Maryland Forceps, Large Needle Driver, Extra Large Needle Driver, Cadiere Forceps, Secure Cadiere Forceps, Double Fenestrated Grasper, and Toothed Grasper. All these instruments can be sterilized and reused for up to three procedures, with the only exception of Monopolar Curved Shears and Needle Drivers, which are currently for single use only. However, these instruments could be designed for more than three uses in the near future. Additionally, Medtronic is planning to integrate LigaSure^TM^ vessel sealing technology into Hugo^TM^ RAS equipment. LigaSure^TM^ is a widely used tool that combines pressure and energy to achieve complete and permanent vessel fusion. The controllers provided by Hugo^TM^ RAS have “pistol-like” handles that offer excellent ergonomics and improved dexterity. This design ensures better control over the tip of the needle, which is crucial for precision and error-free maneuvers. The “trigger” mechanism used for clamping also provides stability to the hand–wrist complex.

During RAPN, efficient communication between the master surgeon and bedside surgical staff is vital. The Hugo^TM^ RAS system features a non-immersive console that allows the master surgeon to communicate effectively with the team. All these elements, from the ergonomic instrument’s design to the advanced console, have the potential to significantly improve intraoperative efficiency, enhance control over unexpected bleeding, and reduce the risk of instrument clashes that could lead to organ injury. It is important to emphasize that during RAPN, coordination between the master surgeon and bed-assistant is pivotal: the Hugo^TM^ RAS independent-cart docking system offers an in vivo adjustable unconstrained docking feature at the bedside, allowing the optimization of angles and tilts without the necessity of altering trocar positions or placing additional ports. These elements contribute to the ability to tailor the surgical strategy to the patient’s characteristics, particularly in complex conditions.

Nevertheless, its application is minimally invasive, whereas urology is actually limited to conventional procedures. However, this system has the potential to expand indications in the future as more surgeons become exposed to this technology and contribute to the evaluation of trocar placement and docking settings. To date, only one series of RAPN with Hugo^TM^ RAS has been published and was comprehensive in ten cT1 renal masses [24]. All surgeries, with the exception of one patient, were completed with on-clamp RAPN using four robotic arms while, in a singular case, suboptimal trocars’ placement caused several clashes between the robotic arms, compromising the safety of the procedure and forcing the surgeon to convert to laparoscopic partial nephrectomy. We propose an alternative three-arm setting with a peculiar configuration tailored to NSS. This is the first study exploring the trocar placement and system docking for NSS, and no clashes between robotic instruments inside the abdomen, nor between robotic arms or with bed-assistant laparoscopic trocars were recorded, as all surgeries were completed without difficulties. The median docking and console time were limited to 5 and 90 min, respectively, while the median EBL was 175 mL. These data are comparable to median values reported by large off-clamp series from tertiary-care centers [25,26,27,28,29,30]. No intraoperative complication occurred as no additional ancillary laparoscopic devices were necessary for improving bleeding control. The identification of anatomical cleavage planes for the preservation of healthy renal parenchyma and the collecting system was achieved through the excellent synergy between the master surgeon and the bed assistant. Noteworthy, three patients (12%) reported post-operative minor complications (Clavien–Dindo 2) requiring intravenous antibiotics injection and a single-unit blood transfusion, respectively.

From the bed assistant’s point of view, the Hugo^TM^ RAS system demonstrated excellent ergonomics. The ability to customize the trocar and arm carts’ configuration separately provided an enhanced working space and helped avoid unexpected system failures due to instrument clashing. As far as we know, this represents the largest series of RAPNs aiming to evaluate the efficacy and safety of a unique trocar and docking setting of the Hugo^TM^ RAS System with an independent three-arm cart configuration. In our experience, this setting allowed for the optimal use of working space, minimizing internal and external instrument clashing.

Notwithstanding the simplified trocar placement, the independent modular docking system, and the promising results obtained with the Hugo^TM^ RAS, it must be recognized that all procedures were performed by a widely experienced team for purely off-clamp minimally invasive partial nephrectomy. Another limitation of the present study is the relatively small sample size of the population, which could impede the generalization of our outcomes without prior external validation. Accordingly, these results might not be universally applicable due to the need for further studies to standardize this peculiar configuration. However, the adoption of Hugo^TM^ RAS for kidney surgery, especially for high-nephrometry score tumors, appears to be encouraging and, due to its reduced procedural costs, could allow more patients to access contemporary high-quality robotic surgery.

## 5. Conclusions

The Hugo^TM^ RAS System demonstrated significant benefits for NSS and, specifically, for off-clamp RAPN using this peculiar three-arm configuration. In experienced hands, this simplified three-instrument arrangement of the Hugo^TM^ RAS System for off-clamp RAPN resulted in feasible and safe practice, providing patient-tailored trocar placement and docking with non-inferior outcomes compared to other robotic platforms. These preliminary results may pave the way for urologists to embrace this novel robotic system for RAPNs and even to investigate its application to complex renal masses.

## Figures and Tables

**Figure 1 jpm-13-01372-f001:**
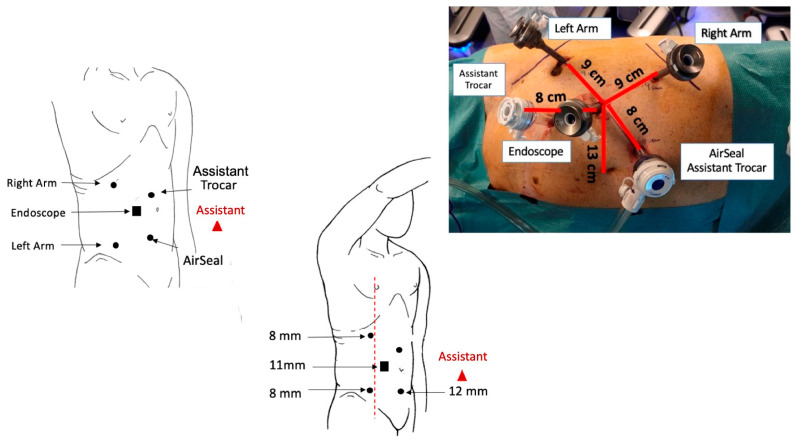
Trocar placement and configuration for right RAPN.

**Figure 2 jpm-13-01372-f002:**
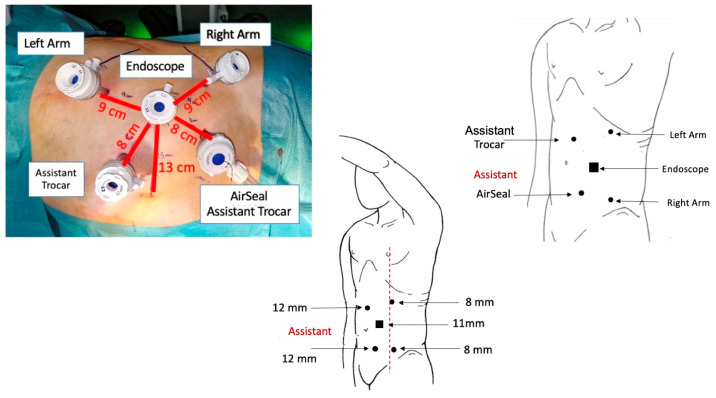
Trocar placement and configuration for left RAPN.

**Figure 3 jpm-13-01372-f003:**
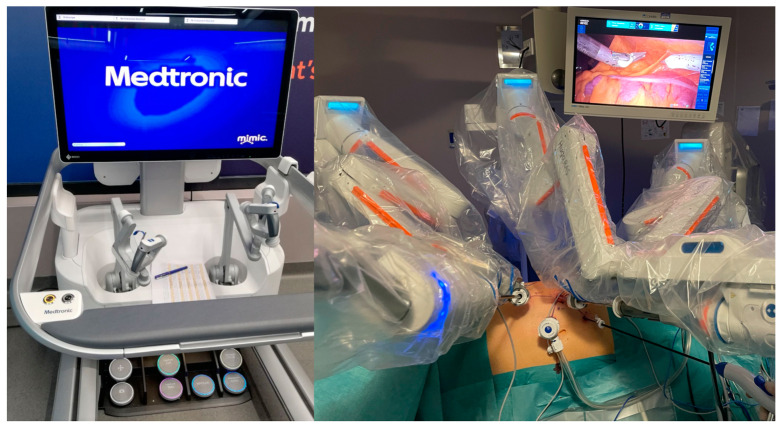
The Hugo^TM^ RAS System console ((**left**) image) and the three arm cart configurations for RAPN ((**right**) image).

**Table 1 jpm-13-01372-t001:** Baseline and demographic data.

Variable	Cohort (n = 25)
Age (n, median, IQR)	69 (60–73)
Gender (n, %)	
*Male*	19 (76%)
*Female*	6 (24%)
BMI (kg/m^2^, median, IQR)	27.3 (25.7–28.1)
ASA score (n, %)	
*I*	1 (4%)
*II*	18 (72%)
*III*	5 (20%)
*IV*	1 (4%)
Charlson Comorbidity Index (median, IQR)	4.5 (3.25–5)
Diabetes (n, %)	3 (12%)
Hypertension (n, %)	14 (56%)
Preoperative Hemoglobin (g/dl, median, IQR)	15 (13.8–15.5)
Preoperative Creatinine (mg/dL, median, IQR)	0.92 (0.81–1.07)
Preoperative eGFR (ml/min/1.73 m^2^, median, IQR)	84.6 (64.5–90.9)
Clinical Tumor Size (mm, median, IQR)	32.5 (26–43.7)
Number of Lesion (n, %)	
1	24 (100%)
2	1 (0%)
cT (n, %)	
*T1a*	19 (76%)
*T1b*	4 (16%)
*T2*	2 (8%)
Side (n, %)	
Right	14 (56%)
Left	11 (44%)
R.E.N.A.L. score (median, IQR)	6 (5–7)

**Table 2 jpm-13-01372-t002:** Perioperative data.

Variable	Cohort (n = 7)
Docking Time (min, median, IQR)	5 (5–6)
Console Time (min, median, IQR)	90 (68–135.75)
Estimated blood loss (ml, median, IQR)	175 (100–400)
Perioperative complications (n, %)	4 (16%)
Length of Stay (days, median, IQR)	3 (3–4)
Hemoglobin at discharge (g/dl, median, IQR)	11.5 (10.2–12.8)
Creatinine at discharge (mg/dL, median, IQR)	0.9 (0.82–1.12)
eGFR at discharge (ml/min/1.73 m^2^, median, IQR)	81.9 (60.6–89.5)
Readmission (n, %)	0 (0%)
Pathological Size (mm, median, IQR)	30 (18.5–40)
Pathology (n, %)	
*Benign*	8 (32%)
*Malignant*	17 (68%)
Histology subtype (n, %)	
*Oncocytoma*	6 (24%)
*Clear Cell*	10 (40%)
*Papillary*	6 (24%)
*Angiomyolipoma*	2 (8%)
*Chromophobe*	1 (4%)
Positive Margins (n, %)	0 (0%)
pT Stage (n, %)	
*1a*	21 (84%)
*1b*	2 (8%)
*2a*	2 (8%)

## Data Availability

The data presented in this study are available on request from the corresponding author.

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
