# Peer review of "Robot-Assisted Renal Surgery with the New Hugo Ras System: Trocar Placement and Docking Settings"

_jpm, 2023, doi:10.3390/jpm13091372_

Round 1

Reviewer 1 Report

Summary

The jpm-2551037 was a very interesting and appealing topic. The authors presented their experience with the novel HUGO RAS System and especially described the trocar placement procedure and docking.

General comments

The grammar and the wording is good enough. There are some grammar and spelling errors that need to be addressed.

Abstract

The abstract is concise, involving all the necessary information about the study.

Introduction

-        On line 59 it is stated that “Partial nephrectomy represents the gold standard….”. The authors could mention up to what size partial nephrectomy should be performed.

Methods

-        Any exclusion criteria should be reported.

-        The authors should mention all the instruments currently available for the HUGO RAS System and if these have monopolar or bipolar energy (line 159). Are the instruments single use or can be sterilized and used again?

-        Do the trocars have to be 8mm for the robotic arms to be placed, or can be smaller?

-        It would be a good idea if a photo of the console unit and the robotic arms could be included.

-        At the “Trocar Placement and System Docking” section it is stated that two assistant trocars are placed. Why are two trocars needed? Can you explain their purpose?

-        At the “Surgical procedure” section it is stated that all surgeries were completed by one experienced surgeon. Does the surgeon have laparoscopic or robotic experience?

-        The assistant is standing or is being seated during the surgery?

Results

-        The results are presented in a very extensive way.

-        The tables and figures are really helpful and necessary for the completion of the authors work.

Discussion

-        The discussion is of great quality and includes updated data.

-        The authors inform the reader about the study limitations. However it could be added as a limitation that the sample size is relatively small.

Conclusion

From the presented data, the conclusion is complete and represents the work that the authors did. However, lines 293-299 (“Its application…docking setting.”) do not constitute a conclusion of the authors’ work and should be modified.

There are some minor grammar and syntax errors throughout the manuscript. A revision should be made.

- Line 27-28 "three instrument configuration"

- Line 80 "instrument clashing" 

- Line 98 "three-arm trocar placement"

(Drop the "s" in these sentences. Same mistake in many occasions throughout the manuscript - Please correct)

-Line 70 "displaying"

Author Response

Referee: 1

Comments to the authors

The jpm-2551037 was a very interesting and appealing topic. The authors presented their experience with the novel HUGO RAS System and especially described the trocar placement procedure and docking.

We thank the reviewer for the comment, the positive feedback, and the time to carefully review the manuscript and to give incisive yet constructive comments, which has greatly helped us improve this revised draft. Our point-by-point reply to each specific comment is below, as follows:

General comments

The grammar and the wording is good enough. There are some grammar and spelling errors that need to be addressed.

The whole manuscript has been revised by a native English mother tongue, grammar and spelling errors addressed, and the article improved accordingly.

Abstract

The abstract is concise, involving all the necessary information about the study.

We thank the reviewer for the positive feedback.

Introduction

-           On line 59 it is stated that “Partial nephrectomy represents the gold standard….”. The authors could mention up to what size partial nephrectomy should be performed.

The manuscript was improved accordingly and an introduction regarding partial nephrectomy indications has been provided according to tumor size. The modifications were highlighted in the manuscript as follow:

According to European Association of Urology (EAU) guidelines, they strongly recommend to offer partial nephrectomy as the treatment of choice to T1 patients (maximum diameter < 7 cm). Nevertheless, T2 patients (> 7 cm) could benefit from partial nephrectomy in case of solitary kidney or chronic kidney disease if technically feasible.

Methods

-        Any exclusion criteria should be reported.

As stated in the methods section, our Institution is an high-volume centre for off-clamp partial nephrectomy. Consequently, all patients eligible for RAPN were included in the study and underwent surgery with the Hugo RAS System. Patients were excluded if not eligible for partial nephrectomy: gross hematuria, not technically feasible, evidence of infiltration on conventional radiological imaging, clinical stage > T2 (cT3-4).

Exclusion criteria were provided and modifications were highlighted in the manuscript as follow:

All patients eligible for RAPN were enrolled in the study. Patients were excluded if displaying contraindications for partial nephrectomy: gross hematuria, not technically feasible, evidence of infiltration on conventional radiological imaging, clinical stage > T2 (cT3-4).

-        The authors should mention all the instruments currently available for the HUGO RAS System and if these have monopolar or bipolar energy (line 159). Are the instruments single use or can be sterilized and used again?

The manuscript was improved accordingly and a specification regarding Hugo RAS instruments available has been added to the main text. The modifications were highlighted in the manuscript as follow:

The Hugo™ RAS System has been equipped by Medtronic with a large variety of robotic instruments with both mono- and bipolar energy, in order to assist surgeons in performing all types of procedures and to adapt the strategy to many surgical scenarios. The 8-mm instruments currently available for the robotic arms are: Monopolar Curved Shears, Bipolar Fenestrated Grasper, Bipolar Maryland Forceps, Large Needle Driver, Extra Large Needle Driver, Cadiere Forceps, Secure Cadiere Forceps, Double Fenestrated Grasper, and Toothed Grasper. All the instruments can be sterilized and reused for up to 3 procedures, with the only exception of Monopolar Curved Shears and Needle Drivers, which are currently for single use. However, this is a temporary situation, as the instruments will be designed for more than 3 uses in the future. Moreover, Medtronic is planning to add to the Hugo RAS equipment the LigaSure™ vessel sealing technology, one of their best and most widely used tools on the market, which delivers a combination of pressure and energy to create complete and permanent vessel fusion.

-        Do the trocars have to be 8mm for the robotic arms to be placed, or can be smaller?

The smallest size of Hugo RAS robotic trocars currently available is 8-mm.

The manuscript was improved accordingly and modifications highlighted as follow:

The Hugo™ RAS is equipped with 11-mm and 8-mm trocars for endoscope and robotic instruments, respectively.

-        It would be a good idea if a photo of the console unit and the robotic arms could be included.

We thank the reviewer for the suggestion, a picture of the console unite along with robotic arms has been added to the main text and caption provided.

Figure 3. The HugoTM RAS System console (left image) and the three arm-carts configuration for RAPN (right image).

-        At the “Trocar Placement and System Docking” section it is stated that two assistant trocars are placed. Why are two trocars needed? Can you explain their purpose?

Two laparoscopic trocars for bed-assistant were placed in order to allow the assistant a more active role during renal tumor enucleation. A specification of their purpose was added to the main text. The manuscript was improved accordingly and modifications highlighted in the manuscript as follow:

The suggested Hugo™ RAS trocar configuration by Medtronic is based on 4 robotic arms, which includes an 11-mm optic port and three 8-mm robotic instruments. This leaves enough space for only one laparoscopic trocar for the bedside assistant. In our surgical setup, we utilize three robotic arms in a three-instrument configuration and have two 12-mm laparoscopic trocars for the bedside assistant. The rationale behind this trocar arrangement is to provide the assistant with a more active role during off-clamp partial nephrectomy. This involvement includes utilizing two surgical suctions with irrigation at same time. The simultaneous suction and irrigation of the resection bed offer clear-cut visualization of tumor borders during enucleation and a precise distinction between healthy renal parenchyma and the renal mass.

-        At the “Surgical procedure” section it is stated that all surgeries were completed by one experienced surgeon. Does the surgeon have laparoscopic or robotic experience?

A specification of first surgeon experience about laparoscopy and robotics has been added to the main text. The manuscript was improved accordingly and modifications highlighted as follow:

For instance, the first surgeon possesses extensive experience of over 10 years and has performed more than 500 off-clamp laparoscopic partial nephrectomies. Additionally, to gain confidence with the robotic system, following the certified training provided by Medtronic, an initial series of 15 robotic radical prostatectomies with the Hugo™ RAS System has been successfully completed.

-        The assistant is standing or is being seated during the surgery?

The manuscript was modified accordingly and changes highlighted as follow:

The bed-side assistant can either stand up or be seated during surgery. This choice depends on the individual patient's anatomical characteristics and, consequently, on the height of the surgical bed, docking, and tilt angles of the robotic arms. On the other hand, the first surgeon needs to be seated during surgery for optimal handling of Hugo™ RAS controllers.

Results

-        The results are presented in a very extensive way. The tables and figures are really helpful and necessary for the completion of the authors work.

We thank the reviewer for the comment and the positive feedback about results section.

Discussion

-        The discussion is of great quality and includes updated data. The authors inform the reader about the study limitations. However it could be added as a limitation that the sample size is relatively small.

We thank the reviewer for the positive comment and suggestion. The limitation section was modified accordingly and changes highlighted as follow:

Another limitation of the present study is the relatively small sample size of the population, which could impede the generalization of our outcomes without prior external validation.

Conclusion

From the presented data, the conclusion is complete and represents the work that the authors did. However, lines 293-299 (“Its application…docking setting.”) do not constitute a conclusion of the authors’ work and should be modified.

We thank the reviewer for the punctual suggestion. Lines 293-299 have been moved to the discussion section and highlighted (Lines 327-332).

Comments on the Quality of English Language

There are some minor grammar and syntax errors throughout the manuscript. A revision should be made.

- Line 27-28 "three instrument configuration"

- Line 80 "instrument clashing"

- Line 98 "three-arm trocar placement"

All the sentences have been corrected accordingly to reviewer’s suggestion and highlighted in the main text. Furthermore, an extensive English review has been performed from native English.

(Drop the "s" in these sentences. Same mistake in many occasions throughout the manuscript - Please correct)

All the “s” of these sentences have been dropped. Corrections have been highlighted in the main text.

-Line 70 "displaying"

The word displaying was spell-checked and highlighted in the main text.

Reviewer 2 Report

The Hugo RAS system seems to be a high-tech new system for minimally invasive surgical system. It is very interesting to see how well it works in patients with renal tumor. According to authors experience it is a superior system. However if data of comparing to laparoscopic surgery or Da Vinci RAS system can be provided, it might be able to explain more. Regardless, this is a well written manuscript describing the Hugo RAS in renal surgery.  

Author Response

Comments to the authors

The Hugo RAS system seems to be a high-tech new system for minimally invasive surgical system. It is very interesting to see how well it works in patients with renal tumor. According to authors experience it is a superior system. However if data of comparing to laparoscopic surgery or Da Vinci RAS system can be provided, it might be able to explain more. Regardless, this is a well written manuscript describing the Hugo RAS in renal surgery. 

We appreciate the positive feedback and the time taken to thoroughly review the manuscript. A comparison of data with laparoscopy or the Da Vinci system could provide valuable information and insights to our readers. Unfortunately, our institution does not have other robotic systems like the Da Vinci for data comparison. However, we are currently conducting an evaluation and comparison of off-clamp partial nephrectomy between laparoscopy and the Hugo RAS system. This upcoming study will constitute a distinct and separate work, to be published independently from this manuscript. Nonetheless, we express our gratitude to the reviewer for their specific comment and helpful suggestion, which will be duly considered for future studies.
